# A cointegration analysis of rabies cases and weather components in Davao City, Philippines from 2006 to 2017

**Zython Paul T. Lachica**[1], **Johanna Marie Peralta**[1], **Eliezer O. Diamante**[1], **Lyre Anni E. Murao**[2], **May Anne E. Mata**[1]*, **Pedro A. Alviola IV**[3]

**1** Department of Mathematics, Physics, and Computer Science, University of the Philippines Mindanao, Davao City, Philippines, **2** Department of Biological Sciences and Environmental Studies, University of the Philippines Mindanao, Davao City, Philippines, **3** School of Management, University of the Philippines Mindanao, Davao City, Philippines

☯ These authors contributed equally to this work.

* memata@up.edu.ph

**Data Availability Statement:** All relevant data are within the manuscript and its Supporting Information files.

## Abstract

Rabies is a lethal viral disease and dogs are the major disease reservoir in the Philippines. Spatio-temporal variations in environmental factors are known to affect disease dynamics. Some rabies-affected countries considered investigating the role of weather components in driving rabies cases and it has helped them to strategize their control efforts. In this study, cointegration analysis was conducted between the monthly reported rabies cases and the weather components, such as temperature and precipitation, to verify the effect of weather components on rabies incidence in Davao City, Philippines. With the Engle-Granger cointegration tests, we found that rabies cases are cointegrated into each of the weather components. It was further validated, using the Granger causality test, that each weather component predicts the rabies cases and not vice versa. Moreover, we performed the Johansen cointegration test to show that the weather components simultaneously affect the number of rabies cases, which allowed us to estimate a vector-error correction model for rabies incidence as a function of temperature and precipitation. Our analyses showed that canine rabies in Davao City was weather-sensitive, which implies that rabies incidence could be projected using established long-run relationship among reported rabies cases, temperature, and precipitation. This study also provides empirical evidence that can guide local health officials in formulating preventive strategies for rabies control and eradication based on weather patterns.

## Introduction

Rabies is an acute viral infection that causes encephalomyelitis [1], which affects the nervous system of infected mammals. Motor weakness, altered sensorium, and other significant morbidity are some of the effects of rabies infection [2]. Infected animals can transmit this virus to body openings of humans such as wounds through their excreted saliva. The majority of

**Funding:** All authors are funded by the Commission on Higher Education of the Philippines with Discovery Applied Research and Extension for Trans/Inter-disciplinary Opportunities (CHED DARE-TO) 2017 Research Grant through the Synoptic Study on Transmission and Optimum Control to Prevent (STOP) Rabies Research Program. The funders had no role in study design, data collection and analysis, decision to publish, or preparation of the manuscript.

**Competing interests:** The authors have declared that no competing interests exist.

rabies-related human deaths in the world are due to dog bites, although vampire bats are the main reservoir in some regions [3]. There have been rare cases of rabies transmission through inhalation [4].

During the early 1930s in the Philippines, 90% of the animal brain specimens that were examined and recorded to be rabies positive in Manila City came from dog samples. The yearly number of rabid dogs from 1930 to 1934 ranged from 14 to 38, with a total of 145 rabies cases [5]. In the late 1950s, the compulsory annual vaccination and the licensing of dogs in Manila were implemented, resulting in a rabies-free city for seven years [5]. In a recent report, about 635 dogs are infected in the Philippines annually since 2012 [6]. In Davao City, a highly urbanized city with 1.632 million inhabitants [7] and the largest city in the country, it was also observed that most of the animal rabies cases are coming from dog samples (with a few reports coming from cat samples), with a total of 210 rabies-infected dogs reported from 2006 to 2017 (CVO 2019, personal communication, 30 January).

The Philippines aims to be rabies-free by 2022 [8]. Control interventions have been implemented in various localities of the country to prevent the spread of rabies virus in animals. In Davao City, mass dog vaccination has been implemented since 2006 and additional control interventions such as impounding and neutering were implemented in 2011 but despite all these, the number of rabies cases continues to fluctuate (CVO 2018, personal communication, 30 July). These pose a challenge in achieving a rabies-free community. According to Tohma and colleagues [9], control interventions are complicated by dog migration between neighboring islands and by the difficulty of achieving the minimum required 70% vaccination coverage. Furthermore, rabies is considered as one of the lowest priority diseases by the local health authorities [10], i.e., a neglected tropical disease.

Natural indicators, e.g. weather conditions, which are related to rabies incidence can provide additional insights for rabies control (e.g., vaccinating all dogs free of charge [11] and dog population control [12]). For instance, in Africa, the dry season appears to be a risk factor for rabies transmission as it was found that dispersal and mating of animals, as well as rabies cases, are more frequent in this season [13]. Moreover, weather-related information has been used in predicting the incidence of several health-related conditions [14]. Several papers have also explored modeling the relationships between weather components and disease incidences [15–17].

If weather conditions (e.g. precipitation and temperature) are natural indicators of rabies cases, it follows therefore that the number of cases can be predicted using these parameters, especially if the two variables have a long-term association with each other. The long-term association between two or more non-stationary time-series data (i.e., mean and variance of time-series vary over time [18]) can be detected by using cointegration techniques. Also, cointegration captures the dynamical relationship of a time-series response variable to its previous observations as well as to other time-series explanatory variables. This approach has potential utility for the development of strategic interventions on rabies control, e.g. determining the appropriate intervention for certain seasons.

However, cointegration techniques are commonly applied and well-explored in the field of economics [19–21], and a few papers demonstrate its use in the context of diseases [22–24]. To our knowledge, cointegration techniques have not yet been explored in analyzing the pattern of rabies incidence concerning weather patterns. Since these weather components are known to be non-stationary (e.g. precipitation and temperature) [25,26], modeling can be done using cointegration.

In this paper, we demonstrate the use of cointegration techniques in studying both the individual and simultaneous long-term relationships of precipitation and temperature with the rabies cases in Davao City, Philippines. The empirical results of this paper can serve as a basis

for review, strategic planning, and integration of weather factors in local health programs related to rabies. Furthermore, the methodology described here is also applicable to other localities or even to other infectious diseases.

## Materials and methods

This section presents the data used in this study and the detailed statistical modeling procedures conducted. Specifically, this section is divided into the following subsections: data, stationary tests, and lag-length selection (i.e., preliminary analysis), the modeling of rabies cases via cointegration techniques, and the post-estimation diagnostics (stability, impulse-response, and forecast-error variance decomposition).

### Data

There are three datasets in this study considering 144 monthly observations. The first dataset is the monthly reported rabies cases in Davao City, Philippines from the years 2006 to 2017, which were collected from the CVO. The other two datasets included the average monthly data for temperature (in degrees Celsius) and the average monthly data for precipitation (in millimeters). These were collected from the Philippine Atmospheric, Geophysical, and Astronomical Services Administration (PAGASA), Davao City [27]. The data before 2006 was not available since the reported records can no longer be tracked by the local health agency.

### Stationarity tests and lag-length selection

For the time-series variables $x_t$ and $y_t$ to be cointegrated, the obtained first-differenced values for each of the time-series should be stationary [28]. The first-differenced values, denoted as $I(1)$, for $x_t$ and $y_t$ are the change between consecutive observations in the original series, as shown below respectively.

$$x_t^{'} = x_t - x_{t-1} \text{ and } y_t^{'} = y_t - y_{t-1} \tag{1}$$

By convention, stationarity tests are performed using the Augmented Dickey-Fuller (ADF) test [29]. The lag-length was specified, based on the Schwarz information criterion (SIC), upon performing the ADF test. Too small lag-length could invalidate the Cointegration test while too large lag-length results to a loss of power of the test [30,31]. Furthermore, the criterion used in selecting the optimal lag-length for Engle-Granger and Johansen cointegration tests is the [32] and HQIC [33] criteria, respectively. Once stationarity is achieved, cointegration tests can be performed. Else, an appropriate model framework should be applied (e.g. autoregressive distributed lag model) [34].

### Modeling rabies cases via cointegration techniques

Cointegration analysis aims to detect the stable long-run relationship among non-stationary variables at level but is stationary at $I(1)$ [28]. If two time-series variables $x_t$ and $y_t$ are stationary at $I(1)$, then both variables share similar stochastic trends and their difference is also stationary. This means that $x_t$ and $y_t$ do not diverge too far from each other in the long run [34]. Therefore, studying the long-term pattern of one time-series variable in terms of the other variable is possible through Cointegration techniques. Furthermore, a sample size of 144 is enough for a Cointegration analysis since it is already greater than the considered fairly large sample size ($n \geq 100$) [35].

In this paper, we used the Engle-Granger cointegration test to determine if the rabies cases are cointegrated to each weather variable namely, temperature and precipitation. Moreover,

the Johansen's cointegration test was used to see if the rabies cases are cointegrated simultaneously to both weather variables. If there is sufficient statistical evidence of cointegration, error-correction models (precipitation and rabies cases and/or temperature and rabies cases) and a vector error-correction model (both weather conditions and rabies cases) are estimated. These models estimate the rate at which the disturbed long-term pattern of rabies cases and weather components return to stability. Also, these models measure the short and long-run effects of the weather components on the changes in the rabies cases. The empirical specification of the error-correction model to represent the dynamic relationship between the rabies cases and a weather component is shown below,

$$\Delta R_t = -\alpha(R_{t-1} - \beta_1 - \beta_2 W_{t-1}) + \delta_0 \Delta W_t + \delta_1 \Delta W_{t-1} + v_t. \qquad (2)$$

In Eq 2, $R_t$ and $W_t$ are the number of rabies case and the measurement of a weather component at month $t$, respectively where $\Delta R_t = R_t - R_{t-1}$ and $\Delta W_t = W_t - W_{t-1}$. The parameter $\alpha > 0$ is the adjustment/correction coefficient for the disturbances in the long-run relationship of the rabies cases and the specific weather components. Parameters $\delta_0$ and $\delta_1$ represent the short-run effects of the weather component. The parameter $\beta_2$ represents the long-run effect of the weather component and $v_t$ is the error term. Similarly, the vector error-correction model representing the dynamical relationship between the rabies cases and multiple weather components is described by,

$$\Delta R_t = \delta_1 \Delta R_{t-1} + \delta_2 \Delta Prec_{t-1} + \delta_3 \Delta Temp_{t-1} - \alpha ECT_{t-1} + u_t, \qquad (3)$$

where $ECT_{t-1} = R_{t-1} - \beta_1 - \beta_2 Prec_{t-1} - \beta_3 Temp_{t-1}$.

In Eq 3, $R_t$ is the number of rabies cases, $Prec_t$ is the amount of precipitation, and $Temp_t$ is the temperature at month $t$, respectively. The parameter $\alpha$ of Eq 3 is the adjustment/correction coefficient for the disturbances in the long-run relationship of the rabies cases to precipitation and temperature, simultaneously. Parameters $\delta_1$, $\delta_2$ and $\delta_3$ represent the short-run effects of the changes in the number of rabies cases, precipitation, and temperature. The parameters $\beta_2$ and $\beta_3$ represent the long-run effects of the amount of precipitation and temperature to the number of rabies cases at month $t$. Eq 2 and Eq 3 were based on the general form of the ECM and VECM of [18].

**Engle-Granger cointegration test.** The Engle-Granger method initially obtains the errors based on the regression model [34]. The dependent and independent variables are regressed to generate the residuals and are tested for the occurrence of unit roots using the ADF test. If the time-series are cointegrated, then the residuals are stationary [34]. The null hypothesis of the Engle-Granger test assumes no cointegration, while its alternative hypothesis assumes that there is cointegration between variables [28]. To reject the null hypothesis, the test statistic/calculated values should be less than the critical value. After testing for evidence of cointegration between variables, a confirmatory test was performed to ensure that the weather variable ($W_{t-1}$) predicts the rabies cases ($R_t$), with four possible situations to consider [36]:

1. $W_{t-1}$ causes $R_t$;

2. $R_{t-1}$ causes $W_t$;

3. there is a bi-directional causality; and

4. $W_{t-1}$ is independent of $R_t$ (no causality).

To address this, the Granger causality test was used to confirm that the weather variable/s ($W_{t-1}$) granger cause/s the rabies cases ($R_t$) and not the other way around [37].

**Johansen's cointegration test.** The Johansen's cointegration test, on the other hand, was used to determine whether or not a single variable is predicted by two or more other variables simultaneously [37]. This test identifies the number of cointegrating variables to the dependent variable. More precisely, the null hypothesis states that there are at most $r$ cointegrating vectors. The test started from the hypothesis that there are no cointegrating vectors in a vector autoregression model, that is, $r = 0$, then $r \leq 1$, and so on [38]. Once the null hypothesis cannot be rejected, the test for the number of cointegrating vectors stops. If the null hypothesis has been rejected for $r \leq k$-1 and is not rejected at $r \leq k$, then the number of cointegrating vectors is $k$. In some cases, trace statistics and maximum eigenvalue statistics may yield different results and indicates that in this case, the outcome of the trace test was chosen [39].

**Post-estimation diagnostics.** Similar to other models, post-estimation diagnostics are helpful to detect potential model problems. In this study, finding the eigenvalue stability condition [18], and generating the impulse response functions (IRF) and forecast-error variance decomposition (FEVD) were performed as post-estimation diagnostic procedures. With the eigenvalue stability condition, the VECM model is said to be stable whenever the modulus of each eigenvalue is less than one [16, 37]. On the other hand, the IRF and FEVD for 10 months of the rabies cases were also obtained to examine the behavior of the response variable to certain impulses (i.e., an unpredictable change) driven by the variables [40]. The impact of the impulse can either be transitory or permanent [18].

## Results

### Temporal patterns of rabies cases and weather conditions

In this paper, the time-series plots of the variables and their first differences are presented in Fig 1 to determine whether or not the time-series satisfies the conditions for cointegration analysis. Three variables namely monthly positive rabies cases, average precipitation (in millimeters), and average temperature (in degree Celsius) were considered. Fig 1 reports the average rabies cases, average precipitation, and average temperature as 1.4653, 5.3629 millimeters, and 28.2674˚C. Since using the visual representation of the time-series is not reliable in determining stationarity, the confirmation is carried out using formal statistical tests.

### Test for stationarity

Stationarity testing for both time-series and first differences were performed using the ADF test. A lag-length of 13 was selected for all time-series variables and their first differences in conducting the test for data stationarity (see S1 Table). The ADF test, as shown in Table 1, revealed that the rabies cases, precipitation, and temperature are all non-stationary at level since the calculated values are all greater than the critical values at 1%, 5%, and 10% level of significance. On the other hand, the first differences of the rabies cases, precipitation, and temperature are all stationary since the calculated values are all less than the critical values at 1%, 5%, and 10% level of significance.

### Engle-Granger cointegration test and Granger causality test

The Engle-Granger test demonstrated cointegration of both precipitation and temperature to rabies cases since the calculated values (-11.17 for precipitation and -11.18 for temperature) were both less than the critical values (between -3.97 to -1.61) at 1%, 5%, and 10% level of significance (Table 2). Furthermore, the directionality of the relationship was validated through the Granger causality test, wherein significant results were obtained ($p < 0.001$) when the

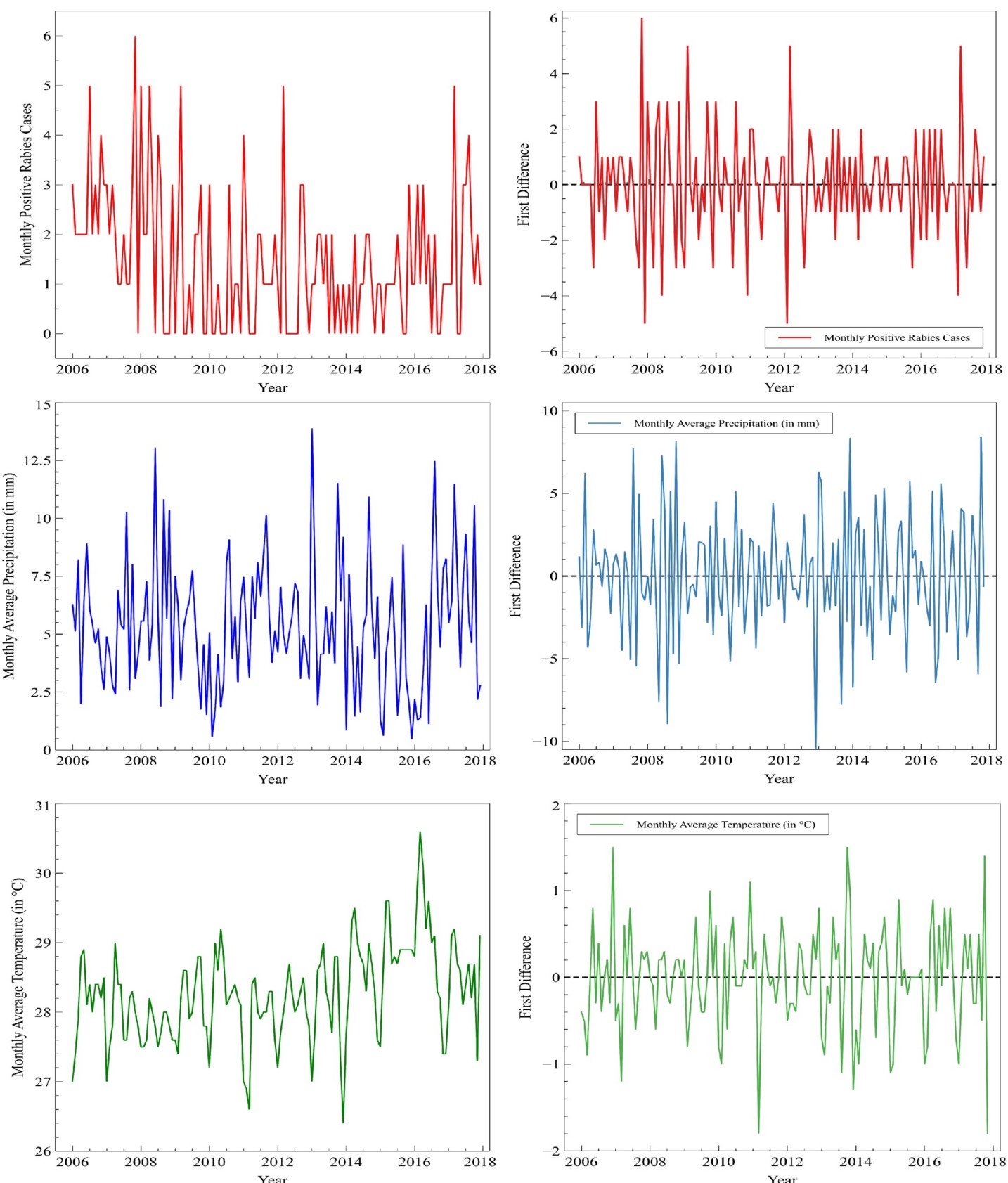

**Fig 1.** The time-series (left) and the first difference (right) plots of the monthly observations of rabies cases, precipitation, and temperature.

**Table 1. Augmented Dickey-Fuller tests for the rabies cases and weather components.**

| Variable | Calculated Value | Critical Value | | | Implication[c] |
|---|---|---|---|---|---|
| | | 1% | 5% | 10% | |
| Number of rabies cases[a] | -1.216 | -2.596 | -1.950 | -1.612 | Nonstationary |
| Number of rabies cases[b] | -5.359 | -2.596 | -1.950 | -1.612 | Stationary |
| Precipitation[a] | -1.718 | -2.595 | -1.950 | -1.613 | Nonstationary |
| Precipitation[b] | -11.839 | -2.596 | -1.950 | -1.613 | Stationary |
| Temperature[a] | 0.402 | -2.596 | -1.950 | -1.612 | Nonstationary |
| Temperature[b] | -2.866 | -2.596 | -1.950 | -1.612 | Stationary |

[a]At-level time-series.

[b]First differences time-series.

[c]$H_0$: Data is nonstationary. $H_1$: Data is stationary. $H_0$ is rejected whenever the calculated value is less than or equal to the critical value.

weather variables are treated as the independent variable and the rabies cases as the dependent variable (Table 3).

**Error-correction models.** Two error-correction models in the form of Eq 2 were estimated in this paper. The estimated coefficients for each model are presented in Table 4 with the corresponding Akaike information criterion (AIC) and Bayesian information criterion (BIC) values: 498.3938 and 504.3055, respectively, when the weather variable is temperature; and 502.9143 and 517.6935, respectively, when the weather variable is precipitation. The statistical estimate for the correction coefficient parameter $\alpha$ is significant at 1% for both models. In addition, the statistical estimate for the constant $\beta_1$ is significant at 1% when the weather variable under consideration is precipitation.

## Johansen's test for cointegration

Based on the HQIC criterion, the optimal lag-length in determining the maximum rank, i.e., the number of cointegrating vector, is 1 (see S2 Table). As shown in Table 5, the maximum rank is 2 since the null hypothesis was rejected at $r = 2$. Hence, there are two cointegrating vectors (i.e. temperature and precipitation) for a VECM with rabies cases as the dependent variable.

**Vector error-correction model.** Table 6 shows the estimated parameters of the reduced form of the VECM for the rabies cases. Both short-run and long-run parameters were included. All the statistical estimates for the long-run parameters were significant at 1%. For the short-run parameters, only the statistical estimate for $\delta_3$, i.e., the short-run effects of the changes in the temperature, was not significant. The VECM had an AIC and BIC values of 10.5094 and 10.8009, respectively, which are relatively small compared to those of the ECMs.

**Table 2. Statistical values derived from the Engle-Granger cointegration of the weather components to rabies cases.**

| Variable | Calculated Value | Critical Value | | | Implication[a] |
|---|---|---|---|---|---|
| | | 1% | 5% | 10% | |
| Precipitation | -11.172 | -3.974 | -3.379 | -3.074 | Cointegrated |
| Temperature | -11.183 | -2.596 | -1.950 | -1.612 | Cointegrated |

[a]$H_0$: Weather component and rabies cases is not cointegrated. $H_0$ is rejected whenever the calculated value is less than or equal to the critical value.

**Table 3. Granger causality test for rabies cases and the weather components.**

| Dependent Variable ($y_t$) | Independent Variable ($x_{t-1}$) | $p$-value[a] |
|---|---|---|
| Rabies cases | Precipitation | p<0.001*** |
| Precipitation | Rabies cases | 0.078 |
| Rabies cases | Temperature | p<0.001*** |
| Temperature | Rabies cases | 0.457 |

[a] If significant:

** $p$-value < 0.05 is significant at 0.05

*** $p$-value < 0.01 is significant at 0.01 where $H_0$ ($x_{t-1}$ does not cause $y_t$) is rejected when $p$-value < 0.05.

## Post-estimation diagnostics

Table 7 shows the eigenvalues computed to evaluate the stability of the model. As seen in the table, out of six eigenvalues, the modulus of two eigenvalues was 1 while the remaining were less than 1.

IRF and FEVD post-diagnostics were performed to reveal how rabies cases respond to the impulse or shocks from the previous rabies cases, precipitation level, and temperature level. As seen in the IRF of Fig 2, the rabies cases quickly respond to the one-time positive shock from the three variables. The IRF also reveals that rabies cases become nearly stable after the first month (from the impulse of precipitation) and after the fourth month (from the impulse of rabies cases and temperature). Note that this observation can be linked to the results of the FEVD (Fig 2) which describes the forecast variance for rabies cases due to shocks from the previous rabies cases, precipitation, and temperature.

## Discussion

We conjectured based on the fluctuations we have seen earlier in the time-series plots of the variables and their first differences that each time-series at level is non-stationary while each corresponding first difference is stationary. These observations were confirmed by the ADF test. Monthly rabies cases, average precipitation (in millimeters), and average temperature (in degree Celsius) are all nonstationary at level while their first differences were all stationary. These stationarity results allow the use of the Engle-Granger cointegration test in examining the long-term association of the weather components and rabies cases.

**Table 4. Estimated parameter values of the error-correction models for rabies cases.**

| ECM Parameters | Parameter Coefficient[a] | |
|---|---|---|
| | $W_t$ is temperature | $W_t$ is precipitation |
| $\alpha$ | 0.9357*** | 0.9337*** |
| $\beta_1$ | 7.8874 | 1.2900*** |
| $\beta_2$ | -0.2276 | 0.0297 |
| $\delta_0$ | -0.1636 | -0.0301 |
| $\delta_1$ | 0.2745 | 0.0025 |
| AIC | 498.3938 | 498.0861 |
| BIC | 504.3055 | 503.9978 |

[a]If significant: * $p$-value < 0.10 is significant at 0.10; ** $p$-value < 0.05 is significant at 0.05

*** $p$-value < 0.01 is significant at 0.01.

**Table 5. Johansen's test for cointegration of rabies cases and weather conditions[a].**

| Maximum Rank | No. of Parameters | LL | Eigenvalue | Trace Statistics | Critical Value |
|---|---|---|---|---|---|
| 0 | 0 | -807.7434 | | 182.6143 | 24.31 |
| 1 | 5 | -761.0272 | 0.4797 | 89.1817 | 12.53 |
| 2 | 8 | -716.4503 | 0.4639 | 0.0281 | 3.84[b] |
| 3 | 9 | -716.4363 | 0.0002 | | |

[a]For lag-length 1, HQIC = 10.2410, for lag-length 2, HQIC = 10.4104, for lag-length 3, HQIC = 10.5202. Hence, lag-length 1 was specified in performing the Johansen's cointegration test using the *vecrank* Stata function.

[b]Maximum rank is 2 since Trace Stat < Critical Value.

The Engle-Granger test revealed that each weather component is cointegrated with rabies cases. Furthermore, the Granger causality test validated that the weather variables are the independent variables, that is, each weather component ($W_{t-1}$) predicts the number of rabies cases ($R_t$) and not vice-versa. The results of the Engle-Granger and Granger causality tests also allowed the estimation of two error correction models wherein rabies cases are independently driven by the weather components. The first model assumes that rabies cases are affected by precipitation [13, 41], while the second one assumes that rabies cases are affected by temperature [42, 43]. From our observations on the fluctuations in the time-series plots and the estimated empirical error correction models, we ask: when will the disturbed long-term relationships of the weather variables and rabies cases become stable? The feedback or adjusted effect parameter $\alpha$ estimates how much disequilibrium is corrected per unit time (month). For temperature, an $\alpha$ of 0.9357 was obtained, which means that 93.57% of the disturbances in the temperature is corrected/adjusted per month. This implies that the disturbed long-run relationship between temperature and rabies cases becomes stable after approximately one month ($|\alpha^{-1}| = 1.069$). On the other hand, the estimated ECM corrects the disturbances in the precipitation at a rate of 0.9337. This means that the disturbed long-run relationship between precipitation and rabies cases becomes stable after approximately one month ($|\alpha^{-1}| = 1.0701$).

On the other hand, the Johansen's test revealed that temperature and precipitation both affect the rabies cases, simultaneously. Hence, a VECM for rabies cases was estimated to simultaneously take into consideration both variables as drivers. The information criteria revealed that the estimated VECM is relatively better compared to the two independent ECMs, suggesting that VECM has less information loss in terms of the long-run relationship of the rabies cases and the weather components and thereby validates the concurrent effect of temperature and precipitation on the number of rabies cases.

The preceding results point out that both temperature and precipitation affect the pattern of rabies cases. From Eq 3, the equation below shows the empirical VECM based on the

**Table 6. Coefficients of the parameters in the VECM for rabies cases in Eq 2[a].**

| Short-run Equation Parameters | Coefficient[b] | Long-run Equation Parameters | Coefficient |
|---|---|---|---|
| $\delta_1$ | -0.3206*** | $\alpha$ | 0.3973*** |
| $\delta_2$ | -0.0753* | $\beta_2$ | -0.6424*** |
| $\delta_3$ | 0.3719 | $\beta_3$ | 0.0706*** |

[a]Information criteria: AIC = 10.5094, BIC = 10.8009.

[b]If significant

* *p*-value < 0.10 is significant at 10%, ** *p*-value < 0.05 is significant at 5%

*** *p*-value < 0.01 is significant at 1%.

**Table 7. The eigenvalues and their corresponding moduli for eigenvalue stability condition post-diagnostics.**

| Eigenvalue | Modulus |
| --- | --- |
| 1 | 1 |
| 1 | 1 |
| -0.3513 + 0.0347$i$ | 0.3530 |
| -0.3513−0.0347$i$ | 0.3530 |
| 0.1259 + 0.1143$i$ | 0.1700 |
| 0.1259 + 0.1143$i$ | 0.1700 |

estimated coefficients of the parameters presented in Table 6:

$$\Delta R_t = -0.3206\Delta R_{t-1} - 0.0753\Delta Prec_{t-1} - 0.3973R_{t-1} - 0.2552Prec_{t-1} + 0.0280Temp_{t-1}. \quad (4)$$

The VECM in Eq (4) has a smaller correction parameter value ($\alpha = 0.3973$) compared to the correction parameter values of the ECMs. The VECM estimates that the disturbed long-term

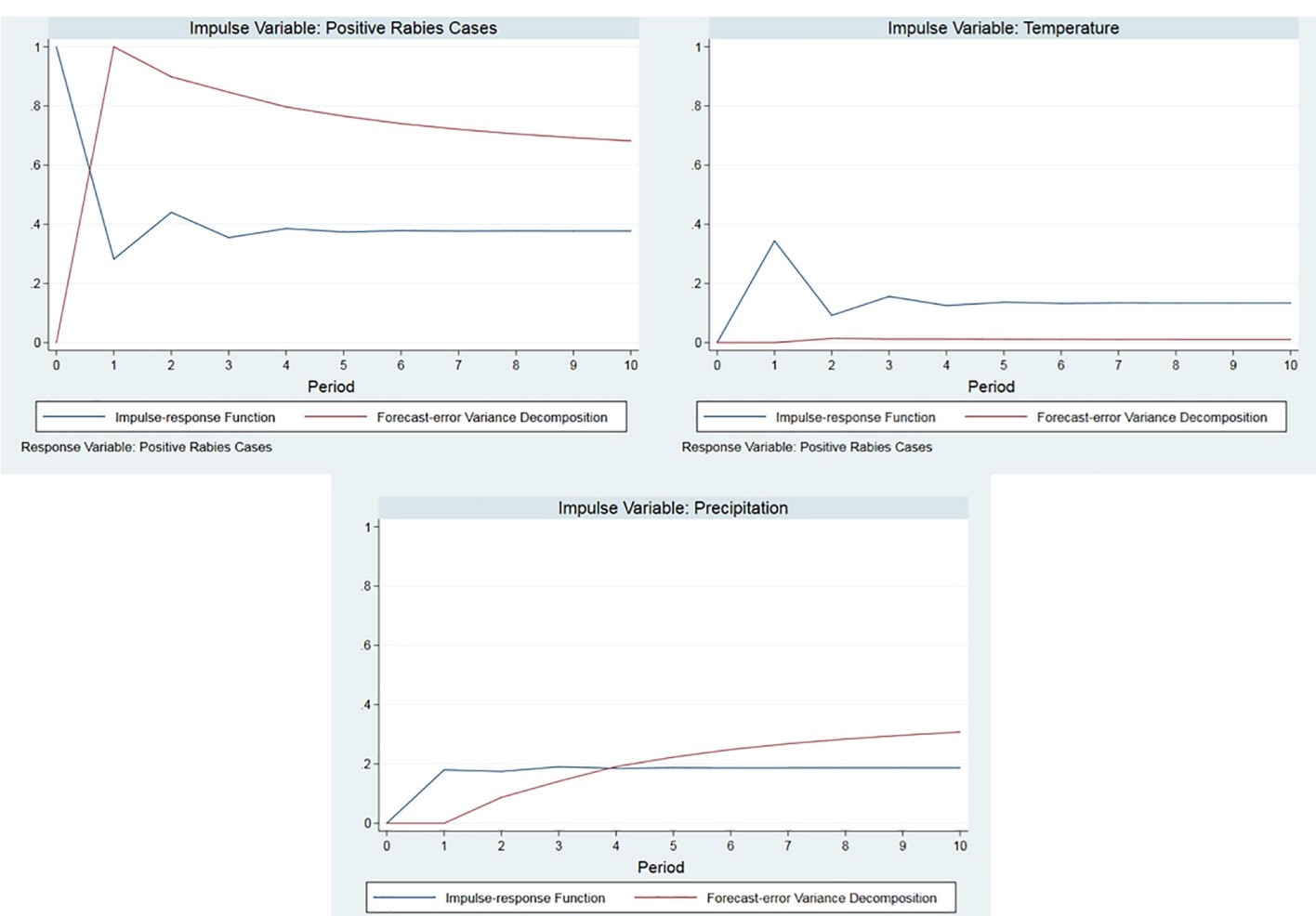

**Fig 2.** Plots of Impulse Response Function (IRF) and Forecast-Error Variance Decomposition (FEVD) of positive rabies cases to impulse on positive rabies cases (top), precipitation (middle), and temperature (bottom) using *Stata 15.1*.

relationship of the rabies cases and weather components takes a longer time, i.e., $|\alpha^{-1}| = 2.5$ months, to go back to the equilibrium state than the estimated time of the ECMs (1 month). Apparently, accounting for the different disturbances of the weather conditions simultaneously will make the feedback or adjusted effect parameter of the VECM relatively lower than with the ECMs. Furthermore, the obtained moduli of the eigenvalues for the estimated VECM coefficients are mostly less than 1 [18, 44]. This implies that the eigenvalue stability condition is satisfied and so the number of cointegrating equations of the estimated VECM in this paper is correctly specified.

It is evident in this study that there is a negative relationship between precipitation and the number of dog rabies cases (i.e., fewer rabies cases during high precipitation). Although this finding has been reported by Lachica and colleagues [17] using count regression analysis, the dynamic relationship of rabies cases and precipitation has been further considered in this analysis. Specifically, we analyzed the short-run and long-run effects of this weather component to the rabies cases. From the long-term estimated parameter coefficient of precipitation in Eq 4, we found that in the long run, an approximate decrease by 1.3686 rabies case per month is expected in Davao City if the precipitation of the preceding month remains on the average at 5.3629 millimeters (i.e., -0.2552 * 5.3629 = -1.3686). According to Kurachi and colleagues [45], rabid dogs are naturally hydrophobic and so the fear of rabid dogs to any form of liquid, despite its furious nature, immobilizes them to move from place to place during rainy periods. Aside from hydrophobia, some dogs develop storm phobia [45]. In an anticipated occurrence of a storm, dogs tend to look for spots wherein storm-related stress is minimal (e.g. loud noises of rain and thunderstorms) [46]. Furthermore, the dispersal and mating season of animals usually occur during the dry season [11], i.e. when precipitation is low, as rain acts as a natural barrier. Thus, during rainy seasons, the possibility of an interaction between a rabid dog to a non-rabid dog decreases [13] thereby slowing down the spread of rabies. Similar findings have been reported for cattle rabies in Costa Rica where rainfall was negatively correlated with rabies incidence in cattle, presumably due to reduced foraging of vampire bats as an effect of rain [47].

This study also revealed that the canine rabies cases will increase when the temperature rises. In the long-run, an approximate increase by 0.7916 rabies case per month is expected in Davao City if the temperature of the preceding month remains on the average at 28.27˚C (i.e., 0.0280 * 28.27 = 0.7916). A similar finding has been observed for human rabies cases in China [48–50]. Increased dog activity on months with high temperatures is a probable cause of the rise in the number of dog bites [50]. Temperature is a potential risk factor of rabies spread as the dog's temper is sensitive to high temperatures [48]. Furthermore, animals are more active and travel with greater distance at warmer temperatures, thereby contributing to rabies spread especially with warmer climate [49].

In this paper, we also studied how the number of rabies cases in the next 10 months respond to various disturbances in the present measurements of weather variables and the number of rabies cases. Using the IRF, it can be observed that the number of rabies cases responds largely to its own shocks relative to the shocks in precipitation and temperature. For instance, a one-time impulse in rabies case by 1 unit leads to a permanent increase in the rabies cases by 0.38 units in the long run. On the other hand, a one-time 1-mm impulse in the precipitation level leads to a permanent increase in the rabies cases by 0.19 units while a 1˚C impulse in the temperature level leads also to a permanent increase in the rabies cases by 0.13 units in the long run. Moreover, the FEVD of the estimated VECM for rabies cases reveals that the shocks in the rabies cases have the largest contribution to the variability of the rabies cases over time. However, it should be noted that the IRFs applied in this study do not account for simultaneous impulses in the weather variables. Hence, the above results should be carefully

interpreted in the light of vector autoregressive (VAR) models, where IRFs are deemed appropriate to use [18]. Unlike with the VAR that the impulse dies out (or termed as transitory), the effect of the impulse for the VECM is permanent as reflected in Fig 2.

The foregoing results imply that, although weather conditions like precipitation and temperature drive the decrease and increase of rabies cases in the long run, respectively, interventions directly targeted to dogs should be intensified to eradicate rabies since the impulse in the present number of rabies cases has the largest contribution to the future number of rabies cases. Nevertheless, precipitation and temperature, being natural indicators of rabies cases, will guide the implementation of control interventions strategically. In the long run, it is strategic to intensify the catching of free-roaming dogs when dogs are more active which usually falls during the summer season. Furthermore, we suggest that the mass dog vaccination and castration campaigns should be conducted before the summer season since we expect that the rabies cases will rise after summer. This targeted vaccination schedule would ensure the dogs' immune protection against rabies even before the onset of the projected peak during the summer season. Also, early castration will keep the dog population from increasing dramatically, thereby minimizing interactions and maintaining herd immunity. Therefore, we recommend the inclusion of these findings in the information campaign materials of the CVO to educate the community about the effects of weather on the future trends of rabies cases. In this way, dog owners will also be properly informed and be able to apply proper precautionary measures on their own volition.

The study is mainly focused in Davao City, Philippines, hence the application of this methodology to other localities in the country may result in unique findings. On a global scale, applying the methodology of this paper to other countries with more than two seasons (e.g. USA [11]) will potentially result in different co-integrating relationships between the rabies cases and weather components. We would also like to point out that the quantitative analysis may be affected by underreporting of rabies cases, which is a major constraint in data collection of notifiable diseases [17]. Finally, all the statistical modeling procedures conducted were based on the protocols of Becketti [18].

## Conclusion

The Philippines' goal is to be rabies-free by 2022. The local government of Davao City has been intensifying campaigns to eliminate rabies with the following strategies: vaccination, castration, impounding, and the conduct of information and education campaign (IEC) sessions. To our knowledge, this is the first report in the country which demonstrates the impact of weather components such as precipitation and temperature on canine rabies incidence via cointegration analysis. Our results show that canine rabies in Davao City is weather-sensitive, therefore projecting the rabies incidence using the established long-run relationship of reported rabies cases and the weather components obtained in the study is possible. These results can be useful in formulating targeted strategies for rabies control based on weather patterns, e.g. intensification of mass dog vaccination prior to the summer season. The analysis of this paper can be further applied to other infectious diseases that are hypothesized to be driven by weather patterns provided that sufficient time-series data for disease incidence and weather variable (e.g. amount of rainfall, the temperature in degree Celsius, etc.) are available.

## Supporting information

**S1 Table. Optimal lag-length selection for the positive rabies cases.**
(DOCX)

**S2 Table. Lag-length selection for Johansen's test for cointegration.**
(DOCX)

## Acknowledgments

We also acknowledge Dr. Gloria N. Marquez, Arlene Lagare, Janice H. Mendoza, and Ma. Noreen J. Eng from the Davao City Veterinarian Office (CVO) for providing the data and assisting in the data collection.

## Author Contributions

**Conceptualization:** May Anne E. Mata, Pedro A. Alviola IV.

**Data curation:** Zython Paul T. Lachica, Johanna Marie Peralta.

**Formal analysis:** Zython Paul T. Lachica, Johanna Marie Peralta, Pedro A. Alviola IV.

**Funding acquisition:** Lyre Anni E. Murao, May Anne E. Mata, Pedro A. Alviola IV.

**Investigation:** Zython Paul T. Lachica, Johanna Marie Peralta.

**Methodology:** Zython Paul T. Lachica, Johanna Marie Peralta, Eliezer O. Diamante, May Anne E. Mata, Pedro A. Alviola IV.

**Project administration:** Lyre Anni E. Murao, May Anne E. Mata.

**Supervision:** May Anne E. Mata, Pedro A. Alviola IV.

**Validation:** Zython Paul T. Lachica, Johanna Marie Peralta, Eliezer O. Diamante, Lyre Anni E. Murao, May Anne E. Mata, Pedro A. Alviola IV.

**Visualization:** Zython Paul T. Lachica, Johanna Marie Peralta, Eliezer O. Diamante.

**Writing – original draft:** Zython Paul T. Lachica, Johanna Marie Peralta, May Anne E. Mata, Pedro A. Alviola IV.

**Writing – review & editing:** Zython Paul T. Lachica, Johanna Marie Peralta, Eliezer O. Diamante, Lyre Anni E. Murao, May Anne E. Mata, Pedro A. Alviola IV.

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
