## [Decision Letter · Decision Letter 0]

14 May 2020

PONE-D-20-04340

A cointegration analysis of rabies cases and weather components in Davao City, Philippines from 2006 to 2017

PLOS ONE

Dear Dr. Mata,

Thank you very much for submitting your manuscript "A cointegration analysis of rabies cases and weather components in Davao City, Philippines from 2006 to 2017" (#PONE-D-20-04340) for review by PLOS ONE. As with all papers submitted to the journal, your manuscript was fully evaluated by academic editor (myself) and by independent peer reviewers. The reviewers appreciated the attention to an important health topic, but they raised substantial concerns about the paper that must be addressed before this manuscript can be accurately assessed for meeting the PLOS ONE criteria. Therefore, if you feel these issues can be adequately addressed, we invite you to submit a revised version of the manuscript that addresses the points raised during the review process. We can’t, of course, promise publication at that time.

We would appreciate receiving your revised manuscript by Jun 28 2020 11:59PM. To enhance the reproducibility of your results, we recommend that if applicable you deposit your laboratory protocols in protocols.io, where a protocol can be assigned its own identifier (DOI) such that it can be cited independently in the future. For instructions see: http://journals.plos.org/plosone/s/submission-guidelines#loc-laboratory-protocols

We look forward to receiving your revised manuscript.

Kind regards,

Abdallah M. Samy, PhD

Academic Editor

PLOS ONE

2. Please address the following:

- Please ensure you have thoroughly discussed any potential limitations of this study within the Discussion section.

- Please refrain from stating p values as 0.000, either report the exact value or employ the format p<0.001.

3. Please upload a copy of Figure 3, to which you refer in your text on page 20. If the figure is no longer to be included as part of the submission please remove all reference to it within the text.

**Reviewers' comments:**

Reviewer's Responses to Questions

**Comments to the Author**

1. Is the manuscript technically sound, and do the data support the conclusions?

Reviewer #1: Yes

Reviewer #2: Yes

2. Has the statistical analysis been performed appropriately and rigorously?

Reviewer #1: Yes

Reviewer #2: Yes

3. Have the authors made all data underlying the findings in their manuscript fully available?

Reviewer #1: Yes

Reviewer #2: Yes

4. Is the manuscript presented in an intelligible fashion and written in standard English?

Reviewer #1: Yes

Reviewer #2: Yes

5. Review Comments to the Author

Reviewer #1: The paper is well written and was able to establish on how weather affects rabies incidence in a particular area in the Philippines. The statistical calculations performed were sound and valid to establish weather's effect on rabies incidence.

Reviewer #2: Rabies is an acute neurological infection of humans and animals, caused by rabies virus and usually transmitted by animal bites. Rabies is a neglected zoonotic disease. Several animal species can transmit rabies, but domestic dogs are the main reservoir implicated in rabies transmission to human and other susceptible animals. The study comes from a good research group from Philippines in surveillance of animal rabies according to Engle-Granger cointegration tests in Davao City, Philippines, the largest city in this country. The results identified that high precipitation acts as a barrier for dog mobility while high temperature promotes dog activity. This environmental study could provides evidence for the role of weather on rabies incidence in Philippines, which can guide decision-makers and health officials to strategize rabies interventions based on this criteria to fight against this major deadly zoonosis dog mediated. However, i have some suggestions to reorganize the paper. At this stage, this paper needs to be much improved. This paper presents results, indeed worthy of publication but not with this form which must be revised. I urge an organization of this paper to be evaluated for the second time.

General comments

- Please use the double space between the different chapters in the entire manuscript.

-Please suppress line numbers for empty lines throughout the manuscript.

- I urge the consultation of author guidelines of the journal for manuscript organization. Author must respect each section (Beginning section, Middle section, Ending section, Other elements).

- Authors must enumerate the different sections and chapters of your manuscript as it is indicated in author instruction to avoid disorder.

1. Abstract:

- Lines 21-22: Use reservoir instead of vector for dogs.

- Authors must include keywords at the end of the abstract.

2. Introduction:

- Line 64: Use reservoir instead of vector for dogs here and throughout the manuscript.

- Lines 72-76: Give the right number of infected rabies dogs in Davao City. Your expertise area.

- Line 83: According to [9] !!!! Give the name of the first author.

- Line 88: (provide additional insights for rabies control). Give reference at least in your neighboring countries.

3. Methods (Use Materials and Methods as presentation of this section)

I urge that all paragraphs of Materials and Methods section will be enumerate.

- Lines 123-124: Put the reference of this sentence.

4. Results and discussion

This section must be divided separately into results and discussion. You have to expose results and after discuss your scoop.

- Line 334: Thanks to compare your res

6. PLOS authors have the option to publish the peer review history of their article (what does this mean?). If published, this will include your full peer review and any attached files.

Reviewer #1: No

Reviewer #2: No

---

## [Author Response · Author response to Decision Letter 0]

25 Jun 2020

REPLY TO THE ACADEMIC EDITOR

First and foremost, we thank you Dr. Abdallah Samy for taking some time to evaluate our manuscript amidst the pandemic. We feel that your comments have substantially improved the paper. Below are our replies to each point you raised:

Authors’ Response: Thank you for pointing this out. Unfortunately, we were having difficulty accessing the links you provided. We followed the instructions for meeting PLOS ONE’s style requirements using these links instead: https://journals.plos.org/plosone/s/file?id=4497%2FMain%20Body%20-%20ONE%20Formatting.pdf and https://journals.plos.org/plosone/s/file?id=7797%2FTitle%20Page%20-%20ONE%20Formatting.pdf. We hope that this is a valid approach and satisfied your request.

2. Please address the following:

- Please ensure you have thoroughly discussed any potential limitations of this study within the Discussion section.

- Please refrain from stating p values as 0.000, either report the exact value or employ the format p<0.001.

Authors’ Response: We appreciate that you suggested these. We included a thorough discussion on potential limitations of the study in the Discussion section (please see Lines 407-414, Page 19-20) as advised. Moreover, we used the format p<0.001 for very small p-values as you recommended. 

3. Please upload a copy of Figure 3, to which you refer in your text on page 20. If the figure is no longer to be included as part of the submission please remove all reference to it within the text.

Authors’ Response: We apologize for this. There was no Figure 3 and it was a typographical error. We replaced Figure 3 with Figure 2, which is the correct reference. We thank you for this comment. 

REPLY TO REVIEWER 1

Thank you very much for taking time in reading the manuscript amidst this COVID-19 crisis. We, the authors, sincerely appreciate all your efforts in reviewing our paper. There are a few formatting changes that we made in the paper to abide by the standards of the journal. Finally, we are grateful to know that you gave a positive response to the manuscript and we will do our best to address any further comments. Below are our replies to each point you raised:

The paper is well written and was able to establish on how weather affects rabies incidence in a particular area in the Philippines. The statistical calculations performed were sound and valid to establish weather's effect on rabies incidence

Authors’ Response: Thank you very much for commending our paper. We appreciate your comments on validating the statistical calculations we performed. 

REPLY TO REVIEWER 2

Thank you very much for the comments on the earlier version of this paper. We acknowledge your efforts in providing insights to our manuscript amidst this crisis. In this version of the paper, we have done our best to address your suggestions especially in reorganizing the paper. Furthermore, we have included/edited some statements and improved the structure of the Results and Discussion section as per your recommendations. Finally, we are very grateful to know that you gave an overall positive response to the paper and the methods introduced and we will strive to fulfill any further comments. Below are our replies to each point you raised:

Rabies is an acute neurological infection of humans and animals, caused by rabies virus and usually transmitted by animal bites. Rabies is a neglected zoonotic disease. Several animal species can transmit rabies, but domestic dogs are the main reservoir implicated in rabies transmission to human and other susceptible animals. The study comes from a good research group from Philippines in surveillance of animal rabies according to Engle-Granger cointegration tests in Davao City, Philippines, the largest city in this country. The results identified that high precipitation acts as a barrier for dog mobility while high temperature promotes dog activity. This environmental study could provides evidence for the role of weather on rabies incidence in Philippines, which can guide decision-makers and health officials to strategize rabies interventions based on this criteria to fight against this major deadly zoonosis dog mediated. However, i have some suggestions to reorganize the paper. At this stage, this paper needs to be much improved. This paper presents results, indeed worthy of publication but not with this form which must be revised. I urge an organization of this paper to be evaluated for the second time.

Authors’ Response: We thank you for your overall comments and sincerity in improving our paper. We also appreciate your recognition that this manuscript is worthy of a publication. We will do our best to reorganize the paper in order to improve the structure of the narrative. 

General comments

- Please use the double space between the different chapters in the entire manuscript.

Authors’ Response: Thank you for your comment. We have edited the format of the manuscript to use the double space in between chapters. 

-Please suppress line numbers for empty lines throughout the manuscript.

Authors’ Response: Thank you very much for your correction. We did our best to remove the line numbers for empty lines throughout the manuscript for better line tracking. However, for the title page, line numbers appear in empty lines to abide with the formatting style instructed by PLOS ONE as found in this link:https://journals.plos.org/plosone/s/file?id=7797%2FTitle%20Page%20-%20ONE%20Formatting.pdf.

- I urge the consultation of author guidelines of the journal for manuscript organization. Author must respect each section (Beginning section, Middle section, Ending section, Other elements).

Authors’ Response: Thank you very much for your recommendation. We have reviewed the Author guidelines of the journal and have incorporated your comments in the manuscript. 

- Authors must enumerate the different sections and chapters of your manuscript as it is indicated in author instruction to avoid disorder.

Authors’ Response: Thank you very much for your insight. We have included lines/paragraph that acts as a roadmap to enumerate the different sections for each chapter. We appreciate your concern in our paper’s organization.

1. Abstract:

- Lines 21-22: Use reservoir instead of vector for dogs.

Authors’ Response: We appreciate your correction. We removed the word “vector” and replaced it with “reservoir”. Please see Line 24, Page 2 in the ‘Abstract’ section.

- Authors must include keywords at the end of the abstract.

Authors’ Response: Thank you very much for this comment and we apologize for not including keywords in the earlier version of the paper. You can now find the keywords at the end of the abstract at Lines 40-41, Page 2.

2. Introduction:

- Line 64: Use reservoir instead of vector for dogs here and throughout the manuscript.

Authors’ Response: Thank you for your correction. We have appropriately replaced the word “vector” to “reservoir”, from this point and throughout the manuscript. For your reference, please see Line 47, Page 3. 

- Lines 72-76: Give the right number of infected rabies dogs in Davao City. Your expertise area.

Authors’ Response: Thank you very much for your suggestion to improve the accuracy of the narrative presented in the paper. We have included a new sentence to indicate the exact number of infected rabid dogs based on our data. Please see Lines 57-58, Page 3 for your reference.

- Line 83: According to [9] !!!! Give the name of the first author.

Authors’ Response: Thank you for this comment and we apologize. We have indicated the name of the first author and properly cited the reference. You can find this correction in Line 64-65, Page 3-4.

- Line 88: (provide additional insights for rabies control). Give reference at least in your neighboring countries.

Authors’ Response: This comment is well-appreciated. We have included additional insights along that line by citing two references on rabies control in our neighboring countries such as Indonesia and Thailand. Kindly see Lines 70-71, Page 4 of the manuscript.

3. Methods (Use Materials and Methods as presentation of this section)

Authors’ Response: Thank you for this comment. We have edited the section title from “Methods” to “Materials and Methods” based on your recommendation. Please see Line 97, Page 5.

I urge that all paragraphs of Materials and Methods section will be enumerate.

Authors’ Response: Thank you very much for your suggestion. An additional paragraph was included to enumerate all the paragraphs of “Materials and Methods” in order to guide the reader. The new paragraph is in Lines 98-102, Page 5 of the manuscript.

- Lines 123-124: Put the reference of this sentence.

Authors’ Response: Thank you for this comment. We have addressed your suggestion and you can find it in Line 110, Page 6 in the manuscript.

4. Results and discussion

This section must be divided separately into results and discussion. You have to expose results and after discuss your scoop.

Authors’ Response: Thank you and we appreciate your valuable insight with regards to the organization of this section. In the new version of the manuscript, the “Results and discussion” section is now divided into two parts: the “Results” section, where we presented all the results of the tests performed, and the “Discussion” section, where we provide insights from our findings. You can find the new Results and Discussion section starting from Line 202 up to Line 414, Pages 10-20. 

- Line 334: Thanks to compare your res

Authors’ Response: This phrase appears to be cut-off and beyond our comprehension. We hope that the reviewer can clarify this for us in the next round of revision, if necessary.

---

## [Editor Report · Decision Letter 1]

6 Jul 2020

A cointegration analysis of rabies cases and weather components in Davao City, Philippines from 2006 to 2017

PONE-D-20-04340R1

Dear Dr. Mata,

We’re pleased to inform you that your manuscript has been judged scientifically suitable for publication and will be formally accepted for publication once it meets all outstanding technical requirements.

Kind regards,

Abdallah M. Samy, PhD

Academic Editor

PLOS ONE

---

## [Editor Report · Acceptance letter]

17 Jul 2020

PONE-D-20-04340R1 

A cointegration analysis of rabies cases and weather components in Davao City, Philippines from 2006 to 2017 

Dear Dr. Mata:

I'm pleased to inform you that your manuscript has been deemed suitable for publication in PLOS ONE. Congratulations! Your manuscript is now with our production department. 

Kind regards, 

on behalf of

Dr. Abdallah M. Samy 

Academic Editor

PLOS ONE